# Association between Periodontitis and Metabolic Syndrome in a Korean Nationally Representative Sample of Adults Aged 35–79 Years

**DOI:** 10.3390/ijerph16162930

**Published:** 2019-08-15

**Authors:** Ji-Soo Kim, Se-Yeon Kim, Min-Ji Byon, Jung-Ha Lee, Seung-Hwa Jeong, Jin-Bom Kim

**Affiliations:** 1Department of Preventive and Community Dentistry, School of Dentistry, Pusan National University, 49 Busandaehak-ro, Mulgeum-eup Yangsan, Gyeongsangnam-do 626-870, Korea; 2BK21 PLUS Project, School of Dentistry, Pusan National University, 49 Busandaehak-ro, Mulgeum-eup Yangsan, Gyeongsangnam-do 626-870, Korea; 3Periodontal Disease Signaling Network Research Center, Pusan National University, 49 Busandaehak-ro, Mulgeum-eup Yangsan, Gyeongsangnam-do 626-870, Korea

**Keywords:** alcohol drinking, HDL-cholesterol, metabolic syndrome, periodontitis, socioeconomic status

## Abstract

This study aimed to evaluate the association between periodontitis and metabolic syndrome (MetS) and to investigate risk factors associated with MetS in Korean adults aged 35 to 79 years. Among individuals aged 35–79 years who participated in the Korea National Health and Nutrition Examination Survey 2013–2015, 8314 participants who completed the required examinations and questionnaires were included. Confounding variables related to demographic and socioeconomic status and systemic and oral health-related behaviors were age, gender, household income, education level, smoking, alcohol intake, physical activity, and frequency of daily toothbrushing. Of the 8314 participants, 32.2% were diagnosed with MetS. The prevalence of MetS was 26.6% and 41.6% in those without and with periodontitis, respectively. Among individuals with periodontitis, the prevalence of MetS was 44.3% in males and 36.9% in females. Compared to non-periodontitis, periodontitis was associated with MetS (adjusted OR = 1.422, 95% CI: 1.26–1.61). Age, frequency of daily toothbrushing, and periodontitis were associated with MetS in both males and females. While current smoking and alcohol intake more than twice a week were significantly associated with MetS in males, household income and education level were significantly associated with MetS in females. The findings suggest that periodontitis can be associated with MetS.

## 1. Introduction

Metabolic syndrome (MetS) is a cluster of risk factors that predispose to cardiovascular disease and is associated with multiple chronic disease states, such as obesity, impaired glucose tolerance, hypertension, and dyslipidemia [1,2,3,4]. The age-adjusted prevalence rates of MetS in Korean adults above 20 years of age were 24.9% in 1998 and 31.3% in 2007 [5]. The age-adjusted prevalence rates of MetS among Korean adults above 20 years of age were 24.9% in 1998 and 31.3% in 2007 [5]. The age-adjusted prevalence of MetS in the U.S. was 24.1% and 27.0% in the National Health and Nutrition Examination Survey (NHANES) III (1988–1994) and NHANES 1999–2000, respectively [6]. There has been an increase in the global prevalence of MetS and it is a high-risk factor for several systemic diseases [7,8].

Several definitions of MetS have been advanced by the World Health Organization (WHO), National Cholesterol Education Program Adult Treatment Panel III (NCEP ATP III), International Diabetes Federation, and the European Group for the Study of Insulin Resistance. However, the definition provided by NCEP ATP III is the most commonly used [9,10,11,12]. While the WHO definition (1998) emphasized insulin resistance as a major risk factor for MetS, the NCEP ATP III definition (2001) included abdominal obesity, which is greatly related with insulin resistance [13]. The NCEP ATP III defining criteria of MetS are the presence of three or more of the following abnormal states: abdominal obesity, hypertriglyceridemia, low High Density Lipoprotein (HDL) cholesterol, hyper-tension, and impaired fasting glucose levels.

Periodontitis is not only a chronic disease, it is also a long-lasting inflammatory disease that is associated with oral bacterial infection and affects up to 90% of the global population [14]. Periodontitis is associated with some systemic conditions including diabetes and obesity [15,16,17,18,19]. Han et al. [20] reported a relationship between periodontitis and MetS; Kim et al. [21] reported that the severe periodontitis could be a risk factor for MetS. Periodontitis and MetS can share a common pathophysiological pathway as they are both related to systemic diseases [22]. Furthermore, Brunner et al. [23] evaluated the effect of social inequality due to employment grade on the prevalence of MetS and reported that the odds ratio for having the MetS, while comparing the lowest with the highest employment grade was 2.2 in men and 2.8 in women. Prescott et al. [24] studied the effect of social gradient due to education level on the prevalence of MetS and revealed that the odds ratio for having the MetS was 0.32 for the highest versus lowest. Although the association between periodontitis and MetS has been validated in the aforementioned studies [20,21], these studies were conducted in some regional populations. Few studies have been reported using Korean nationwide representative samples now. With a strong association between periodontal disease and MetS, it would be necessary to investigate the risk of Mets in Korean adults aged 35 to 79 years with periodontitis, and to evaluate socioeconomic factors, systemic and oral health conditions, and health behaviors that may be associated with MetS. 

A study that evaluated the rate of periodontal destruction before age 40 reported that without interventions, periodontal lesions often progress relatively rapidly [25]. Another study that evaluated the progress rate of periodontitis in a population that had never been exposed to any interventions related to prevention of oral disease showed that tooth loss commenced after age 30 and increased throughout the decade [26]. Han et al. [27] reported that the age threshold for moderate and severe periodontitis was 43 years in men and 49 years in women. It is very important to identify the association between oral health and systemic diseases among adults after age 30 years. This study aimed to evaluate the association between periodontitis and metabolic syndrome (MetS) and to investigate risk factors associated with MetS in Korean adults aged 35 to 79 years.

## 2. Materials and Methods

### 2.1. Study Population and Design

This study utilized data from the Sixth Korea National Health and Nutrition Examination Survey (KNHANES) conducted by Korea Centers for Disease Control and Prevention (KCDC) from 2013 to 2015. The goal of this survey was to gather national data about the health status, health awareness and behaviors, as well as the nutritional intake of South Korea citizens. The sampling method of this survey included a complex, stratified, multistage, probability-cluster survey of a large representative sample. Based on the 2010 Population and Housing Census conducted by Statistics Korea, 11,520 households and 22,948 members of households older than 1 year were selected from 576 geographical units. Response rates of physical examination and health questionnaire were 74.9% in 2013, 73.9% in 2014 and 73.4% in 2015, and response rates of oral health examination were 70.5% in 2013, 66.8% in 2014 and 65.6% in 2015. All participants provided written informed consent. Among the 22,948 individuals who participated in the KNHANES (2013–2015), 13,681 participants were aged 35 to 79. After excluding participants without systemic and oral examinations and with no response to all questionnaires related to health status, 8314 participants were included (Figure 1). 

All subjects gave their informed consent for inclusion before they participated in the study. The study was conducted in accordance with the Declaration of Helsinki, and the protocol was approved by the Ethics Committee of Institutional Review Board of KCDC (2013-07CON-03-4C, 2013-12EXP-03-5C). Since 2015, the KNHANES has been exempted from review following the Bioethics and Safety Act.

### 2.2. Assessment of Periodontitis

The community periodontal index (CPI) was used to measure periodontitis in the KNHANES, and the oral cavity was divided into the following sextants: # 18–14, # 13–23. # 24–28, # 34–38, # 33–43, and # 44–48. CPI was selected as a tool for evaluating the periodontal health status of population by the World Health Organization [28]. The CPI was rated on a scale of 0 to 4:0 = normal; 1 = gingivitis with bleeding on probing; 2 = presence of calculus; 3 = 4–5 mm of probing depth (PD); 4 = 6 mm or more of PD. CPI grades 1 to 2 were defined as absence of periodontitis and CPI grades 3 to 4 were defined as periodontitis. If CPI grades 3 or 4 were present in at least one of the sextants examined for periodontitis, the participant was regarded as a patient with periodontitis. Periodontal assessment was conducted by calibration-trained dentists using a CPI probe (WHO CPI probe, Osung, Korea). The measurement of clinical attachment loss, pocket depth and periodontal treatment status were not performed in the KNHANES (2013–2015).

### 2.3. Assessment of MetS

The assessment of MetS was based on the following five components: (1) abdominal obesity (waist circumference of ≥90 cm for males and ≥85 cm for females); (2) hypertriglyceridemia (serum triglyceride >150 mg/dL or specific treatment for this lipid abnormality); (3) low HDL-cholesterol (<40 mg/dL for males and <50 mg/dL for females or specific treatment for this lipid abnormality); (4) high blood pressure (systolic ≥130 mmHg and diastolic ≥85 mmHg or treatment with antihypertensive agents); (5) fasting serum glucose (≥100 mg/dL or current use of antidiabetic medication). A participant was considered to have MetS when three or more of these five components were present.

### 2.4. Assessment of Confounders

Demographic and socioeconomic variables and systemic and oral health-related behaviors were selected as confounders, including gender, age, household income, education level, frequency of daily toothbrushing, smoking, alcohol intake, and physical activity. Household income was divided into quartiles. Education levels were divided into four groups: below elementary school, middle school, high school, and college or higher. Based on the response to questionnaires, smoking status was divided as never, past, and current smokers. Alcohol intake level was divided into three groups: (1) those who drink less than once a month; (2) those who drink 1–4 times a month; and (3) those who drink more than twice a week. Physical activity (number of days with at least 10 minutes of sustained walking in the previous week) was divided into two groups: (1) less than 4 days and (2) more than 4 days. The frequency of daily toothbrushing was quantified as the number of tooth brushings performed the previous day.

### 2.5. Statistical Analyses

A complex statistical method was used in the analysis of KNHANES data. A complex sample crosstab was created to evaluate the distribution of demographic and socioeconomic variables, systemic and oral health-related behaviors, and the five components of MetS. In addition, complex sample multivariable logistic regression analysis was performed to assess the association between periodontitis and MetS and to investigate risk factors for MetS according to gender. The odds ratio (OR), 95% confidence interval (CI), and *p* values, by multivariable logistic regression, were estimated. Statistical analysis was conducted using the Statistical Package for the Social Sciences (SPSS), version 25.0 (IBM SPSS Statistics for Windows, Armonk, NY, USA), and statistical significance was set as *p* < 0.05.

## 3. Results

Out of 8314 participants, 37.0% were diagnosed with periodontitis. Moreover, the prevalence of periodontitis by gender was as follows: 45.3% in males and 28.0% in females. The prevalence of MetS associated with periodontitis was as follows: 54.3% in males and 38.3% in females (Table 1). 

Among 8314 participants, 32.2% were diagnosed with MetS. Participants aged 75–79 years, with ≤elementary school education, with lowest quartile of household income, who were current smokers, with alcohol intake of more than twice a week, with 0–1-times daily toothbrushing, and with periodontitis were associated with MetS. The prevalence of MetS was 37.0% in males and 27.0% in females. The proportion of non-periodontitis associated with MetS was 30.9% in males and 23.1% in females, and the proportion of periodontitis associated with MetS was 44.3% in males and 36.9% in females (Table 2).

Compared to non-periodontitis, periodontitis was associated with MetS (OR: 1.422, 95% CI: 1.26–1.61). Moreover, age (OR: 1.041, 95% CI: 1.04–1.05), gender (OR: 1.264, 95% CI: 1.06–1.51), education level (≤ elementary; OR: 1.520, 95% CI: 1.23–1.88, middle; OR: 1.320, 95% CI: 1.08–1.62), current smoker (OR: 1.351, 95% CI: 1.11–1.64), alcohol intake of more than twice a week (OR: 1.245, 95% CI: 1.07–1.45), and frequency of daily toothbrushing (OR: 0.887, 95% CI: 0.84–0.94) showed statistical significance. Males showed statistical significance in age (OR: 1.023, 95% CI: 1.02–1.03), current smoking (OR: 1.252, 95% CI: 1.00–1.57), alcohol intake of more than twice a month (OR: 1.564, 95% CI: 1.27–1.92), frequency of daily toothbrushing (OR: 0.866, 95% CI: 0.80–0.94), and periodontitis (OR: 1.534, 95% CI: 1.31–1.80). Females showed significance in age (OR: 1.068, 95% CI: 1.06–1.08), upper middle quartile of household income (OR: 1.288, 95% CI: 1.04–1.60), education level (≤elementary; OR: 2.231, 95% CI: 1.63–3.05, middle; OR: 2.044, 95% CI: 1.51–2.77, high; OR: 1.297, 95% CI: 1.02–1.66), frequency of daily toothbrushing (OR: 0.911, 95% CI: 0.83–1.00), and periodontitis (OR: 1.325, 95% CI: 1.11–1.58) (Table 3).

## 4. Discussion

We investigated the association between periodontitis and MetS among Korean adults aged 35–79 years using data from the Sixth KNHANES (2013–2015) and verified the risk factors for MetS. We found that periodontitis could be a risk factor for MetS and that MetS was more common in males than in females. 

Periodontitis shares pathological features with the systemic pathologies [29]. Visceral adipose tissue is an important organ that secretes various bioactive substances known as adipocytokines, including tumor necrosis factor-α, and tumor necrosis factor-α secreted from adipose tissue may affects periodontal tissue directly [30,31]. Periodontitis is more common disease in diabetes patients, and worsens with diabetes [32]. The development of impaired glucose tolerance was associated with periodontitis [33,34]. Furthermore, Peroxisome proliferator-activated receptors produced in periodontal inflammation could be one of the meeting background points with atherosclerosis, cardiovascular disease, diabetes and metabolic syndrome [35].

MetS is becoming highly prevalent in developing and developed countries [3,36]. In Korea, the prevalence of MetS increased every year, by 0.6%, over 10 years [5]. In addition, among 15,540 Chinese adults aged 35–74 years surveyed in 2000–2001, the prevalence of MetS was 9.8% in males and 17.8% in females [36]. These findings may be attributed to the impact of rapid economic growth and introduction of westernized diets [36,37]. Since MetS is a familiar metabolic disorder caused by the increasing prevalence of obesity, fundamental preventive approaches, such as weight reduction and raise in physical activity, can serve as potent interventions to this global epidemic [38].

Regular physical activity not only improves physical strength but also reduces obesity and diseases such as cardiovascular diseases and diabetes [39,40]. Park et al. [41] researched the effect of weight decrease on MetS, and showed that a moderate decrease of weight occasioned significant decrease of systolic and diastolic blood pressure, triglyceride, waist circumference. In addition, Kim et al. [42] have studied the associations of regular walking and body mass index on MetS, and reported that physical activity and weight reduction may decrease the incidence of MetS. However, in this study, the association between MetS and physical activity was not statistically significant. This may be due to the varied nature of the intensity and frequency of physical activity and the characteristics of participants. In future studies, it is imperative to clarify the number of days and type of physical activity.

Freiberg et al. [43], who researched the relationship between alcohol intake and prevalence of MetS in the U.S., reported that mild-to-moderate alcohol intake was related with a lower MetS prevalence. Furthermore, Yoon et al. [44] showed that 1–15 g of alcohol intake per day was related with decreased MetS prevalence. In this study, alcohol intake was investigated to be a risk factor for MetS. These findings contradict that of previous studies [43,44]. This could be due to differences in the amount of alcohol consumed. When converting the amount of alcohol consumed more than twice a week to that consumed per day, greater intake might be uncovered. In addition, comparison with previous studies was limited by the lack of data on the amount of alcohol consumed in one sitting. Hence, future studies should consider the amount of alcohol consumed.

Smoking as a risk factor for MetS has been recognized in several studies [45,46]. Ishizaka et al. [47] reported that past and current smoking status was associated with an increased incidence of MetS with odds ratios of 1.77 and 2.38, respectively. This study showed statistical significance for only males. Compared to males, the smoking rate of females in Korea was very low, and females did not show statistical significance (males: past 42.3%, current 40.6%, females: past 4.9%, current 4.9%) (smoking rates by gender are not shown in the table). There may be different study findings in countries with different smoking cultures.

In this study, age, frequency of daily toothbrushing, and periodontitis were associated with MetS, in males and females. While current smoking and alcohol intake more than twice a week were significantly associated with MetS in males, household income and education level were significantly associated with MetS in females. The findings of this study showed that periodontitis, as well as socioeconomic status and oral health behaviors such as frequency of daily toothbrushing, smoking and alcohol intake, could be risk factors for MetS.

Our study has some limitations. Firstly, the information on socioeconomic status and health behaviors was only collected through questionnaires, and therefore, may contain inappropriate information. However, the KNHANES is composed of a nationally representative sample from the Korean population, and the data were extracted using systematic sampling that was adjusted for the number of household members while accounting for the type of residence, administration district, and region in South Korea [48]. Secondly, this research was a type of cross-sectional study; therefore, it could not be used to determine cause-effect relationships between periodontitis and MetS. Further, well-designed, prospective studies will be needed to discover the causal relationships. Lastly, the CPI, utilized to assess periodontal status, is likely to be overestimated or underestimated, because the mouth is partially assessed [49]. However, the CPI is widely used to evaluate periodontitis in collaborations between the WHO and World Dental Federation, and has been validated through various studies [28,50]. Notwithstanding these limitations, this study showed associations between periodontitis and MetS in Korean adults aged 35 to 79 years and found several risk factors associated with MetS.

## 5. Conclusions

This study evaluated the association between periodontitis and MetS and investigated the risk factors related to MetS. The results showed that periodontitis can be a risk factor related to MetS, and age, household income, education level, smoking, alcohol intake, and frequency of daily toothbrushing were associated with MetS. Oral health professionals in Korea should recognize the association between periodontitis and MetS, and the risk factors related to MetS.

## Figures and Tables

**Figure 1 ijerph-16-02930-f001:**
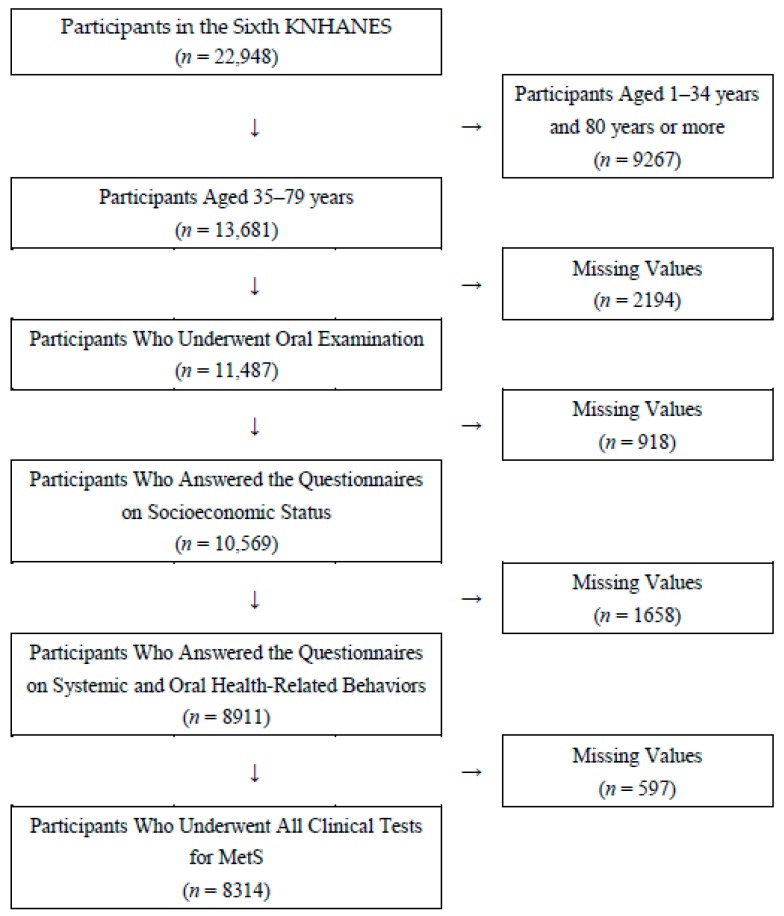
Flow chart for sampling of study participants. MetS: metabolic syndrome.

**Table 1 ijerph-16-02930-t001:** The prevalence of periodontitis according to five components of MetS in total and by sex.

Variables	*n*	Total		Males (3860)		Females (4454)	
		%	*p*-Value *	%	*p*-Value *	%	*p*-Value *
All	8314	37.0	<0.001	45.3	<0.001	28.0	<0.001
Waist circumference							
<90 (male), 85 (female)	6058	34.2	<0.001	43.3	<0.001	24.9	<0.001
>90 (male), 85 (female)	2256	45.1		50.1		38.4	
Fasting serum glucose							
<100	5084	31.6	<0.001	40.4	<0.001	24.2	<0.001
≥100	3230	45.9		51.0		37.3	
Hypertriglyceridemia							
<150	5110	32.4	<0.001	40.6	<0.001	25.9	<0.001
≥150	3204	44.3		50.4		33.3	
Low HDL-Cholesterol							
>40 (male), 50 (female)	4865	35.0	<0.001	42.3	<0.001	24.9	<0.001
<40 (male), 50 (female)	3449	40.3		51.7		31.7	
Blood pressure							
Systolic <130, diastolic <85	4743	31.1	<0.001	39.6	<0.001	23.8	<0.001
Systolic ≥130, diastolic ≥85	3571	46.0		51.8		36.9	
MetS							
less than three	5480	31.9	<0.001	40.0	<0.001	24.2	<0.001
three or more	2834	47.9		54.3		38.3	

MetS: metabolic syndrome; * *p*-values are evaluated by chi-square tests.

**Table 2 ijerph-16-02930-t002:** The prevalence of MetS according to demographic and socioeconomic status, health behaviors, and periodontitis in total and by sex.

Variables	*n*	Total		Males (3860)		Females (4454)	
		%	*p*-Value *	%	*p*-Value *	%	*p*-Value *
All	8314	32.2	<0.001	37.0	<0.001	27.0	<0.001
Age							
35–44	2297	19.8	<0.001	28.8	<0.001	10.1	<0.001
45–54	2158	28.9		35.3		21.9	
55–64	2054	42.1		45.3		38.4	
65–74	1435	52.0		45.4		59.7	
75–79	370	54.8		45.6		64.5	
Education level							
≤Elementary	1805	50.0	<0.001	44.8	<0.001	53.3	<0.001
Middle	1065	40.2		42.7		37.8	
High	2843	28.5		37.1		19.3	
≥College	2601	24.6		32.7		12.3	
Household income							
Lowest quartile	1360	45.5	<0.001	46.0	0.001	45.0	<0.001
Lower middle quartile	2074	33.1		36.8		29.3	
Upper middle quartile	2350	29.8		34.2		24.9	
Highest quartile	2530	28.2		36.4		18.6	
Smoking							
Never	4710	27.5	<0.001	30.6	0.004	26.9	0.847
Past	1976	36.9		37.9		27.8	
Current	1628	37.7		38.6		28.8	
Alcohol intake							
Less than once a month	3359	31.1	<0.001	32.7	<0.001	30.4	<0.001
1–4 times a month	2870	28.4		32.8		23.6	
≥twice a week	2085	38.6		42.9		22.4	
Physical activity							
Less than 4 days	4099	32.7	0.387	37.4	0.593	27.4	0.580
More than 4 days	4215	31.7		36.5		26.6	
Frequency of daily toothbrushing							
0–1	826	43.8	<0.001	45.8	<0.001	38.6	<0.001
≥2	7488	31.0		35.6		26.3	
Periodontitis							
No	5216	26.6	<0.001	30.9	<0.001	23.1	<0.001
Yes	3098	41.6		44.3		36.9	

MetS: metabolic syndrome; * *p*-values are evaluated by chi-square tests.

**Table 3 ijerph-16-02930-t003:** The adjusted association of MetS with periodontitis in total and by sex.

Variables	Total			Males			Females		
	OR ^†^	95% CI ^‡^		OR ^†^	95% CI ^‡^		OR ^†^	95% CI ^‡^	
Age	1.041	1.04	1.05	1.023	1.02	1.03	1.068	1.06	1.08
Sex (Ref. Female)	1.264	1.06	1.51						
Household income									
Lowest quartile	1.052	0.87	1.27	1.060	0.80	1.40	1.091	0.85	1.40
Lower middle quartile	0.948	0.82	1.10	0.870	0.70	1.08	1.164	0.93	1.46
Upper middle quartile	1.015	0.87	1.18	0.885	0.72	1.10	1.288	1.04	1.60
Highest quartile	1			1			1		
Education level									
≤Elementary	1.520	1.23	1.88	0.912	0.69	1.21	2.231	1.63	3.05
Middle	1.320	1.08	1.62	1.039	0.79	1.37	2.044	1.51	2.77
High	1.040	0.90	1.21	1.009	0.83	1.23	1.297	1.02	1.66
≥College	1			1			1		
Smoking									
Current	1.351	1.11	1.64	1.252	1.00	1.57	1.396	0.92	2.12
Past	1.102	0.91	1.34	1.180	0.95	1.47	1.356	0.89	2.06
Never	1			1			1		
Alcohol intake									
1–4 times a month	0.982	0.86	1.12	1.126	0.91	1.40	0.968	0.80	1.17
≥twice a week	1.245	1.07	1.45	1.564	1.27	1.92	0.847	0.64	1.11
Less than once a month	1			1			1		
Physical activity (Ref. ≥4 days)	1.008	0.90	1.13	1.024	0.87	1.20	0.971	0.82	1.15
Frequency of daily toothbrushing	0.887	0.84	0.94	0.866	0.80	0.94	0.911	0.83	1.00
Periodontitis (Ref. No)	1.422	1.26	1.61	1.534	1.31	1.80	1.325	1.11	1.58

Dependent variable: MetS (Yes); ^†^ Odds ratio; ^‡^ 95% CI = 95% Confidenc Interval; MetS: metabolic syndrome; Odds Ratios and 95% Confidence Interval were evaluated by complex sample multivariable logistic regression analysis.

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
