# Peer review of "Association between Periodontitis and Metabolic Syndrome in a Korean Nationally Representative Sample of Adults Aged 35–79 Years"

_ijerph, 2019, doi:10.3390/ijerph16162930_

Round 1
Reviewer 1 Report
This manuscript is a study aiming to determine the association between MetS and periodontitis.
This research has been well conducted, however i did not see the originality of the work. Therefore, i suggest some additions/modifications:
as you mentioned in the introduction, several studies already demonstrated this association. What is the novelty of your study? I agree that yours is perform in the Korean population but, if this is the novelty, can we generalize your conclusions to a larger population? in your method, you determine the diagnosis of perio based on CPI. This description is not accurate and more precise diagnosis will be interest: generalized vs localized pattern?, aggressive vs chronic (from Armitage 1999); stage/grade (CHicago 2017)? ; date of onset?; treatments? same for MetS the statistical analysis has been well conducted, however, implementation of the aforementioned parameters will be of intrest and will strengthen the hypothesis also, develop the cofounding factors role in the discussion sectionAuthor Response
We sincerely thank to you for your careful reading and valuable comments. As you can see, I addressed all comments and paid attention to the important fact. In response to your each comment, we revised our manuscript. Please see the attachment. We sincerely look forward to be given a positive response from you toward a publication of our manuscript.

Reviewer 2 Report
[Suggestions]
4. Discussion
L. 157-159: "The findings from this study, which show the association between periodontitis and MetS, were similar to those of previous studies [20, 21, 28]."
What does it mean? If so, why did the authors perform the present study?
The referee would rather say that the manuscript (particularly, in the Introduction) needs to express rationales of the present study, e.g., Why did the authors evaluate the association between periodontitis and metabolic syndrome?, and Why did the authors investigate risk factors associated with metabolic syndrome in Korean adults aged 35 to 79 years?
Minor:
5. Conclusions
L. 211-213: "Oral health professionals should recognize the association between periodontitis and MetS, and the risk factors related to MetS."
"Oral health professionals" may better be "Oral health professionals in Korea".
Author Response
We sincerely thank to you for your careful reading and valuable comments. As you can see, I addressed all comments and paid attention to the important fact. In response to your each comment, we revised our manuscript. Please see the attachment. We sincerely look forward to be given a positive response from you toward a publication of our manuscript.

Reviewer 3 Report
This article is interesting, and delves into lines studied in relation to periodontitis and other diseases. The article, however, should be improved in the discussion part, since it is not addressed what causal hypotheses could be behind this association, as well as the possible studies that should be proposed to validate these hypotheses are not discussed. Both aspects must be incorporated.
Author Response

(The authors gave the same response as above.)

Round 2
Reviewer 1 Report
Authors addressed all the comments.
Reviewer 2 Report
I recognized and agreed with the revised version of the manuscript.